# Neurophysiological Effects of Virtual Reality Multitask Training in Cardiac Surgery Patients: A Study with Standardized Low-Resolution Electromagnetic Tomography (sLORETA)

**DOI:** 10.3390/biomedicines13071755

**Published:** 2025-07-18

**Authors:** Irina Tarasova, Olga Trubnikova, Darya Kupriyanova, Irina Kukhareva, Anastasia Sosnina

**Affiliations:** Department of Clinical Cardiology, Research Institute for Complex Issues of Cardiovascular Diseases, Academician LS Barbarash Blvd., 6, 650002 Kemerovo, Russia; truboa@kemcardio.ru (O.T.); kuprds@kemcardio.ru (D.K.); kuchin@kemcardio.ru (I.K.); mamoas@kemcardio.ru (A.S.)

**Keywords:** virtual reality, multitask training, postoperative cognitive dysfunction, coronary surgery

## Abstract

**Background:** Digital technologies offer innovative opportunities for recovering and maintaining intellectual and mental health. The use of a multitask approach that combines motor component with various cognitive tasks in a virtual environment can optimize cognitive and physical functions and improve the quality of life of cardiac surgery patients. This study aimed to localize current sources of theta and alpha power in patients who have undergone virtual multitask training (VMT) and a control group in the early postoperative period of coronary artery bypass grafting (CABG). **Methods:** A total of 100 male CABG patients (mean age, 62.7 ± 7.62 years) were allocated to the VMT group (*n* = 50) or to the control group (*n* = 50). EEG was recorded in the eyes-closed resting state at baseline (2–3 days before CABG) and after VMT course or approximately 11–12 days after CABG (the control group). Power EEG analysis was conducted and frequency-domain standardized low-resolution tomography (sLORETA) was used to assess the effect of VMT on brain activity. **Results:** After VMT, patients demonstrated a significantly higher density of alpha-rhythm (7–9 Hz) current sources (t > −4.18; *p* < 0.026) in Brodmann area 30, parahippocampal, and limbic system structures compared to preoperative data. In contrast, the control group had a marked elevation in the density of theta-rhythm (3–5 Hz) current sources (t > −3.98; *p* < 0.017) in parieto-occipital areas in comparison to preoperative values. **Conclusions:** Virtual reality-based multitask training stimulated brain regions associated with spatial orientation and memory encoding. The findings of this study highlight the importance of neural mechanisms underlying the effectiveness of multitask interventions and will be useful for designing and conducting future studies involving VR multitask training.

## 1. Introduction

Cutting-edge medical technologies and their prevalence and availability have contributed to a rise in human life expectancy. However, age is a significant risk factor for cognitive impairment, which leads to disability and a deterioration in quality of life [1]. Besides aging, cardiovascular diseases can cause a decline in cognitive functions [2]. Cardiac surgery is a frequent necessity for treatment of cardiovascular diseases, but it contributes to ischemic brain damage and cognitive decline as well [3,4]. Postoperative cognitive dysfunction (POCD) after cardiac surgery is associated with prolonged stay in the intensive care unit and hospital, deterioration of rehabilitation, and, ultimately, a reduction in the effectiveness of surgery [4].

Currently, there is a lack of complete agreement regarding the ideal approach to prevention of POCD and cognitive rehabilitation in cardiac surgery patients. In the study conducted by M. Butz et al., postoperative cognitive training involved paper-and-pencil exercises. Three-month follow-up results demonstrated that participants who received training demonstrated improved scores in tests for depression and the SF-36 Mental Component Summary [5]. The other study demonstrated that a cognitive training program, which included both memory and attention exercises administered through a computer-based platform, may enhance cognitive function in individuals aged 65 years and older who have undergone coronary artery bypass grafting (CABG). The training sessions were conducted between the sixth and tenth week following the surgery [6]. Moreover, a meta-analysis of 1335 patients who underwent coronary surgery found that those who received cognitive training were at a significantly lower risk of developing POCD. These patients demonstrated significantly improved cognitive function and a higher quality of life [7]. However, these cognitive training techniques impact only one cognitive domain (either attention or memory). Furthermore, the researchers suggest employing these methods no earlier than 1.5 months after surgery. Thus, the authors miss a critical period when postoperative cognitive deficits could regress quickly.

The development of new strategies for cognitive rehabilitation is necessary to preserve quality of life, mental health, and social status after cardiac surgery. Multimodal interventions, which integrate cognitive training, physical activity, and lifestyle changes, are gaining support. A combination of cognitive and physical training provides synergistic benefits for cognitive and physical well-being [8]. Potentially beneficial non-pharmacological strategy for cognitive enhancement and rehabilitation involves utilizing virtual reality (VR) settings. Recent studies have demonstrated the effectiveness of the use of VR approaches in traumatic brain injury and pain management [9,10]. The use of VR-based cognitive rehabilitation programs has the potential to enhance cognitive indicators in mild cognitive impairment and Alzheimer’s disease [11,12]. The previously proposed VRADA-VR Exercise App for dementia and Alzheimer’s patients is one of the first to implement a multitask approach in the virtual environment with cycling and one-digit mathematics. This program helps to restore cognitive functions in patients with Alzheimer’s disease [11,13]. The use of VR technologies also improves cognitive functions by increasing neuroplasticity and cognitive flexibility [14]. A recent meta-analysis has shown that cognitive multitask interventions can activate neuroplasticity mechanisms associated with changes in functional connectivity, including reconfiguring the default mode network and improving the efficiency of fronto-parietal networks. Changes in oscillatory processes, especially increased synchronization in the alpha range, may also be related to cognitive training [8].

The application of VR technologies for cognitive rehabilitation of cardiac surgery patients requires thorough, fundamental research and identification of the most effective technologies for supporting mental well-being and recovering cognitive abilities. VR technologies in cardiac rehabilitation have been discussed in two main categories: physical and psychological rehabilitation [15]. The use of VR for physical rehabilitation has shown positive outcomes, including a reduction in pain and length of hospital stay, and an increase in metabolic equivalents. Moreover, it has positive effects on psychological rehabilitation in terms of decreasing stress, emotional tension, depression, anxiety, and depression scores.

Ischemic brain damage associated with cardiopulmonary bypass and other surgical factors is known to have a multifocal localization [16,17]. We hypothesized that using a rehabilitation approach that involves multitask training with simultaneous performance of motor and cognitive tasks in a virtual environment can be effective in cardiac surgery patients. In addition to that, we hypothesized that VR multitasking enhances cognitive resources by engaging multiple brain regions simultaneously. This increased cognitive load may lead to activation of neuroplasticity processes and restructuring of compensatory brain resources.

At the same time, understanding the mechanisms of plasticity of brain functional systems will deepen with the help of studying the patterns of brain organization associated with the process of cognitive recovery. Digital electroencephalography (EEG) is a widely used non-invasive method for monitoring brain activity and studying the fundamental mechanisms of brain function [18]. According to the study by Tan and colleagues, the combination of VR and EEG technology provides an efficient cognitive rehabilitation training program for patients with mild cognitive impairments [19]. The study by Sadeghi et al. showed that patients with Parkinson’s disease had greater EEG activation during postural stability training compared to healthy controls [20]. In other studies, event-related potentials (ERPs) have been used to verify the effects of VR rehabilitation on brain activity. Specifically, the amplitude of the N1 ERP peak and the amplitudes of alpha, theta, and beta waves are important for understanding changes in brain activity associated with cognitive rehabilitation using VR games [21,22].

Special algorithms related to the solution of the inverse problem allow us to obtain information about generators of the electrical activity of the brain recorded from the scalp surface using EEG [23] and thereby increase the relatively low spatial resolution of the method [24]. These facts prompted us to apply the standardized low-resolution brain electromagnetic tomography (sLORETA) method to evaluate the local activity of both superficial and deep cortical structures of the brain. We used sLORETA in 64-channel EEG to localize the effect of VR-based multitask training (VMT) on brain activity and compared it to the control group; both groups comprised cardiac surgery patients.

## 2. Materials and Methods

### 2.1. Data Collection and Sampling

One hundred male CABG patients (mean age, 62.7 ± 7.62 years) participated in the study. All participants were selected from the cohort of patients of the Research Institute for Complex Issues of Cardiovascular Diseases who were scheduled for elective CABG. We treated all subjects in compliance with the Declaration of Helsinki (revised in 2013). The study was conducted with the full consent of each participant using a protocol approved by the Institutional Review Board of the Research Institute for Complex Issues of Cardiovascular Diseases (protocol No. 5, dated 16 May 2023). The collection of patient data started on January 2024. The inclusion criteria were as follows: stable coronary artery disease (CAD), elective CABG, aged 45–75 years, and provided informed consent. The exclusion criteria were history of stroke, epilepsy, traumatic brain injury, depression, or dementia; score of 18 or lower on the Montreal Cognitive Assessment Scale (MoCA); Beck’s Depression Inventory (BDI-II) score of 8 or higher; or non-cardiovascular decompensated comorbidities. Only male participants were included in the study to eliminate the effect of sex differences on clinical and demographic characteristics and EEG data. All of the patients had normal or corrected-to-normal vision (none of the participants were color-blind).

The overview of the study’s design is provided in Figure 1.

Patients were allocated to the VMT group: (*n* = 50) or to the control group (*n* = 50). We used a pseudo-randomization method to form two groups, depending on baseline clinical and anamnestic characteristics (Table 1).

In all patient groups, elective CABG was performed using normothermic, non-pulsatile cardiopulmonary bypass. Mean cardiopulmonary bypass time was 78.6 ± 20.93 min in the VMT group and 89.4 ± 34.17 min in the control group (*p* ≥ 0.05). Standard techniques for endotracheal anesthesia and drug administration were employed. Continuous monitoring of cerebral cortex oxygenation (rSO2) was performed using an INVOS-3100 device (Somanetics, Troy, MI, USA). Throughout the surgical procedure, oxygen saturation levels remained stable and within acceptable limits. Following CABG, all patients were admitted to the intensive care unit for one to two days. Subsequently, they were transferred to the cardiology department for ongoing care and were discharged from the hospital approximately 11 to 12 days later.

### 2.2. Neuropsychological Examination

In order to include VMT and control patients in the study, we used Russian-modified versions of the Montreal Cognitive Assessment (MoCA) and Beck’s Depression Inventory (BDI-II). In the second stage (2–3 days before CABG), extended neuropsychological testing (attention, short-term memory, psychomotor speed and executive functions) was conducted. POCD assessment was carried out within 2–3 days and 11–12 days after the CABG. POCD was determined for each patient individually, using the percentage of relative changes in postoperative indicators compared to baseline in the following formula: ((baseline value–postoperative value)/baseline value) × 100%. Negative values indicated an increase in the cognitive indicator compared to the baseline, positive values indicated a decrease, and the threshold value for cognitive decline was set at 20% [25]. The examiners were standardized and blinded to the patients’ participation in the study.

### 2.3. Virtual Reality-Based Multitask Training Paradigm

All participants enrolled in the study exhibited POCD at 2–3 days after CABG, as determined by the previously outlined criteria [25]. The VMT course started 3–4 days after CABG, and the number of the daily training sessions depended on the length of patient’s stay at the hospital (approximately 5–9 training sessions). The mean number of training sessions was 6.7. A qualified professional conducted the VMT sessions. The patients were unaware of the task used in the training course. The training specialist gave the patients instructions about the training tasks before the VMT session. We considered VMT successful if POCD was no longer present at follow-up.

A VR display head HP Reverb G2 (HP, Palo Alto, CA, USA) was used to immerse patients in three-dimensional space. A Hori Racing Wheel Apex (Hori, Torrance, CA, USA) was used for the motor task. In a three-dimensional environment, the patient drove a tractor and observed a road stretching along fruit trees in front of them. The patient maintained the vehicle’s trajectory within the road by turning the wheel (motor task) and moving at a low speed (5–15 km/h). The speed of the tractor varied within the specified limits during the training course, depending on the patient’s tolerance. During the tractor’s movement, the patient was instructed to count the number of green and red apples separately (target signals), while ignoring yellow apples (non-target signals). The colored apples appeared on the tree where the tractor was passing. Then, 30 s after the signals appeared on the tree, the patient was asked to select an answer on the internal screen in the form of a number of target signals and press a key on the wheel: the right one if the answer is correct and the left one if it is incorrect. After the training session was finished, the screen displayed the user’s results: correct and incorrect answers, reaction time, and visual encouragement [26].

### 2.4. EEG Data Recording and Processing

All patients underwent resting-state EEG recording with closed eyes. The EEG data of the VMT patients and the control group were acquired via a NEUVO-64 system (Compumedics, El Paso, TX, USA), with a sampling frequency of 1000 Hz and electrode impedances all less than 5 kQ [18]. The electrode at the tip of the nose was used as a reference for recording, while a ground electrode was positioned in the center of the forehead. Data were preprocessed through Neuroscan 4.5 software program (Compumedics, El Paso, TX, USA) and EEGLab toolbox v14.1.2 on MATLAB R2013b (Mathworks Inc., Natick, MA, USA). CB1 and CB2 channels were removed as they were not needed in the study. EEG data were filtered using a bandpass between 1 and 50 Hz. Bad leads were interpolated and bad segments were rejected. The oculographic and myographic artifacts were removed using semi-automatic algorithms (Neuroscan 4.5). Artifact-free EEG segments were divided into two-second epochs. We used 30 epochs for analysis.

Electrical source imaging was performed on the individual frequency bands defined as theta (3–5 Hz) and alpha (7–9 Hz) using sLORETA on the LORETA-KEY software package v20171101. The individual frequency bands were defined by the frequency of the alpha peak in the VMT and control groups (9.3 Hz and 9.4 Hz, respectively) [27]. The selection of these frequency ranges was based on literature data. Thus, theta and alpha oscillations were the most reliable indicators of background EEG activity related to cognitive impairments [28].

The current source density (CSD) was calculated in nanoamperes per square meter (nA/m^2^) at the probabilistic locations, based on the average MRI atlas from the Montreal Neurological Institute (MNI) [23,29]. The neural currents at the source points are modeled as equivalent current dipoles (ECDs) that represent the dominant component of the local current as a vector that has both a magnitude and a direction. Each source point uses three separate dipoles, arranged in three orthogonal orientations, representing the magnitude of the current in the x-, y-, and z-directions [30]. Thus, three orthogonal dipole moments (x, y, and z) were defined and calculated for each source voxel. The source space was restricted to 6239 cortical voxels (5 mm). Only the cortical gray matter and hippocampus were included in the solution space. The dynamic cross-spectrum was calculated using a continuous Gaussian window with a width of 440 ms. Following that, we estimated the current density within the analyzed frequency ranges for each of the 6239 voxels.

### 2.5. Statistical Analysis

The clinical and demographic data were analyzed using Statistica 10.0 software (StatSoft, Tulsa, OK, USA, SN: BXXR210F562022FA-A). The parameters are presented as the mean with SD and the number of observations (*n*, %). The distribution of variables was assessed by the Shapiro–Wilk test. Most of the clinical variables were non-normally distributed. Continuous variables were evaluated using Mann–Wallis one-way analysis of variance and Wilcoxon tests. Continuity-corrected χ^2^ tests were used to determine categorical variables and the percentage relative change in postoperative indicators. Statistical analysis of the indicators of CSD was performed using the sLORETA package and the statistical non-parametric mapping method (SnPM), with 5000 randomizations and log-transformed data [23]. The differences were considered significant at *p* < 0.05.

## 3. Results

### 3.1. Source Estimation Analysis in VMT Group

The success of the VMT program was achieved in 46% of patients (23 out of 50) with absence of POCD at 11–12 days after CABG. The behavioral results for successful and unsuccessful groups are presented in Table 2.

Source estimation analysis revealed no significant changes in theta rhythm (3–5 Hz) in all patients with VMT (t > −3.68; *p* < 0.19).

There was a significant activation of alpha-rhythm (7–9 Hz) current sources (t > −4.19; *p* < 0.01) in Brodmann area 30, parahippocampal, and limbic system structures as compared to preoperative data (Figure 2).

MNI coordinates for brain areas showing the largest CSD differences between the preoperative and postoperative data are presented in Table 3.

### 3.2. Source Estimation Analysis in Control Group

Source estimation analysis revealed a significant increase in theta rhythm (3–5 Hz) in the control patients at 11–12 days after CABG (t > −3.98; *p* < 0.017). There was a notable postoperative increase in the current source of the theta rhythm (3–5 Hz) within the occipital lobe, cuneus, and posterior cingulate regions compared to baseline (Figure 3A). MNI coordinates for brain areas showing the largest CSD differences between the preoperative and postoperative data in the control group are presented in Table 4.

In addition, the control group showed a significant increase in alpha rhythm (7–9 Hz) at 11–12 days after CABG (t > −3.92; *p* < 0.03). The occipital lobe, cuneus, and middle occipital gyrus appear to have a higher source density than before surgery (Figure 3B). Table 4 shows the brain areas with the five largest differences in CSD for the alpha-frequency band.

### 3.3. Contrast Between VMT and Control Groups

We analyzed the differences between the control and VMT groups in terms of training effectiveness. A preliminary statistical analysis of the preoperative and postoperative data showed that CSDs for theta- and alpha-frequency bands were not significantly different between all VMT patients and the control group before and after CABG.

Next, we compared the successful VMT group with the control group (*n* = 23). The analysis revealed that the successful VMT group had significantly lower theta and alpha CSDs in the frontal lobe as compared to the control group (t > 3.64; *p* < 0.01 and t > 3.61; *p* < 0.02, respectively) (Figure 4A,B). Table 5 shows the brain areas with the five largest differences in CSD.

## 4. Discussion

This study aimed to evaluate the effects of VR multitask training on brain activity in cardiac surgery patients. A significant finding of the study was an increase in theta-band CSDs after CABG compared to baseline in the control group. We did not see such increase in patients who underwent VMT. Previously, it has been shown that there is an increase in the theta rhythm in states of emotional arousal, fatigue, and drowsiness [28,31,32]. A recent study also found that individuals with brain damage who exhibit fatigue when performing balance tasks in a 3D environment also demonstrate an increased power of the theta rhythm [32]. The power of theta activity increases in patients after cardiac surgery and subjects with neurodegenerative diseases [18,28], which can be interpreted as a sign of brain dysfunction. Certain brain disorders may manifest a specific oscillatory pattern, known as thalamocortical dysrhythmia [33]. In cardiac surgery patients, an increase in theta activity may be associated with cerebral ischemia during cardiopulmonary bypass. This cerebrovascular insufficiency can cause neuronal dysfunction, tissue atrophy, and damage to neural networks, leading to cortical suppression by subcortical regions and a dominance of low-frequency brain activity [18,34]. Our data suggested that VMT was not linked to an increase in theta power in CABG patients. The activation of compensatory mechanisms in the brain during VMT might be the cause of such a positive outcome.

Our findings also demonstrated that the organization of sources of background cortical activity in VMT patients was associated with alpha activity. We noted an increase in the density of neural sources in the limbic system and the parahippocampal area in patients with VMT. The amygdala and the hippocampus/parahippocampal regions are key components of the limbic system that play a critical role in emotional learning and memory [35]. In the study by Crivelli and Balconi [36], source analysis revealed a higher alpha current density during periods of internally directed attention, with the main cortical generator in the right parahippocampal gyrus. The parahippocampal area became active while performing the task of searching for landmark objects in VR space [37]. Thus, the patients with VMT had an increase in alpha-range oscillatory activity in the areas related to spatial memory and internalized attention.

Alpha oscillations represent a neural mechanism that facilitates communication between different regions of the brain [38]. Previous studies using the EEG method to estimate brain correlations of cognitive load reported that multitasking activity can lead to an increase in alpha power [26,39]. The presence of higher alpha-wave activity was associated with the state of monotony [40]. Additionally, it has been established that increased alpha-band activity inhibits information processing in non-task-related modalities [41]. We suppose that the increase in alpha brain activity in areas responsible for spatial memory and internalized attention in VMT patients can be used as an indication of adaptive mechanisms occurring during training in a three-dimensional environment.

The obtained intergroup differences indicated that patients with successful VMT had a lower density of theta and alpha neural activity in the frontal lobe (middle frontal gyrus, superior frontal gyrus) compared with the control group. The functional significance of these brain regions is linked to sustained attention and spontaneous cognition, as well as alertness during task performance, which supports the maintenance of cognitive function [42]. The middle frontal gyrus is a part of the ventral and dorsal attention systems, as well as the fronto-parietal network, and plays a crucial role in top-down attentional control [42,43,44]. The behavioral results of our study demonstrated that patients with successful VMT showed positive changes in attentional and short-term memory indicators. As previously discussed, recent studies have demonstrated an increase in resting-state theta-wave activity, which is associated with thalamic–cortical dysregulation [28,33]. At the same time, increased alpha synchronization may be related to the deactivation of cortical regulatory processes [45]. The course of VMT can trigger reparative processes within the brain that would change the oscillatory activity in the frontal cortex.

It should be noted that the findings of the LORETA analysis were obtained in a pilot study. Specifically, the results highlighting the differences between the successful VMT patients and controls were shown using a smaller sample (23 patients), indicating lower reliability of these findings.

Currently, there are not enough studies focusing on how cognitive training in 3D spaces activates neuroplastic processes. Understanding these mechanisms is crucial for improving rehabilitation strategies for patients undergoing cardiac surgery and developing effective methods to restore cognitive function. The negative consequences of brain injury in patients who have undergone coronary surgery still persist for an extended period of time [46]. Based on the data analyzed in this study, we can conclude that implementing VR multitask training during the initial recovery phase after CABG has a moderate level of effectiveness and positively influences the rearrangements in neurophysiological processes.

## 5. Limitations

When interpreting the results of our study, it is important to take into account some limitations. Firstly, the effectiveness of VMT for patients was assessed through individual analysis of their cognitive performance before and after surgery. Moreover, VMT was performed during a short early postoperative period after CABG. Secondly, the study had a small sample size, as we were able to recruit a limited number of consecutive patients. Finally, to better understand the long-term effects of VR training on cognitive performance, studies should be conducted one year after the completion of the training program. Consequently, further research is needed to successfully adapt multitask training methods for the virtual reality interface in order to effectively train memory, executive functions, and attention.

## 6. Conclusions

The organization of the sources of background cortical activity in CABG patients after a VMT course was associated with an increase in CSDs in the alpha band in the limbic system and parahippocampal region. Thus, the virtual reality-based multitask training stimulated brain regions associated with spatial orientation and memory encoding. The control group showed an increase in theta-band CSDs after CABG compared to baseline, but this increase was not seen in patients who had undergone VMT. The findings of this study highlight the importance of neural mechanisms underlying the effectiveness of multitask interventions and will be useful for designing and conducting future studies on multitask training.

## Figures and Tables

**Figure 1 biomedicines-13-01755-f001:**
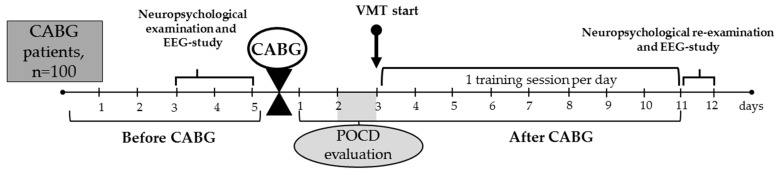
Overview of the study. CABG, coronary artery bypass grafting; EEG, electroencephalography; POCD, postoperative cognitive dysfunction; VMT, virtual multitask training.

**Figure 2 biomedicines-13-01755-f002:**
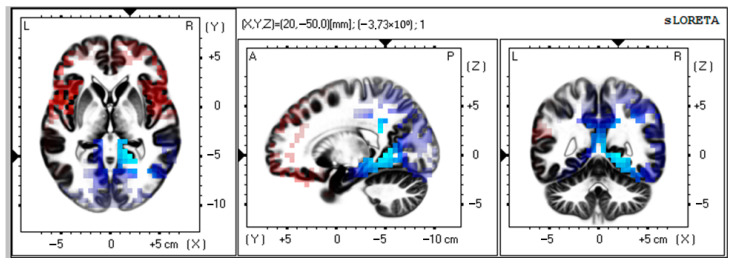
Source localization of alpha activity in VMT group (*n* = 50). Blue blobs indicate that activity before surgery was higher than after the training course and vice versa for red blobs.

**Figure 3 biomedicines-13-01755-f003:**
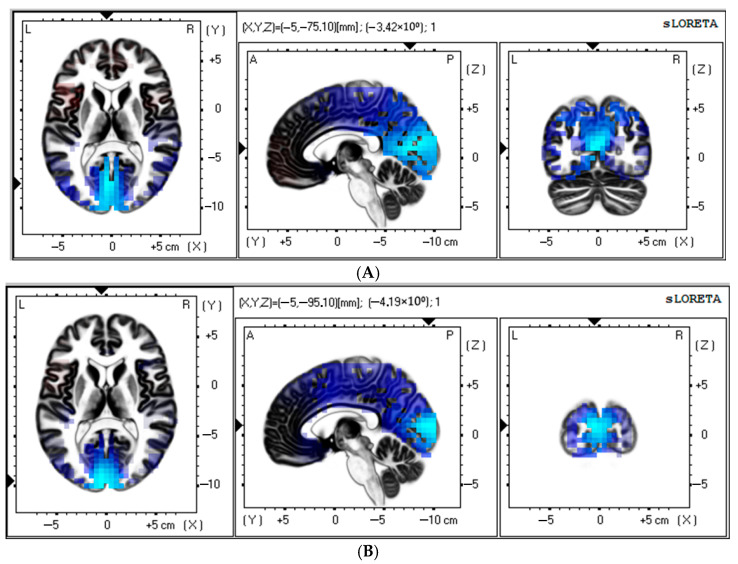
Source localization of theta (**A**) and alpha (**B**) activity in the control group (*n* = 50). Blue blobs indicate that activity before surgery was higher than after CABG.

**Figure 4 biomedicines-13-01755-f004:**
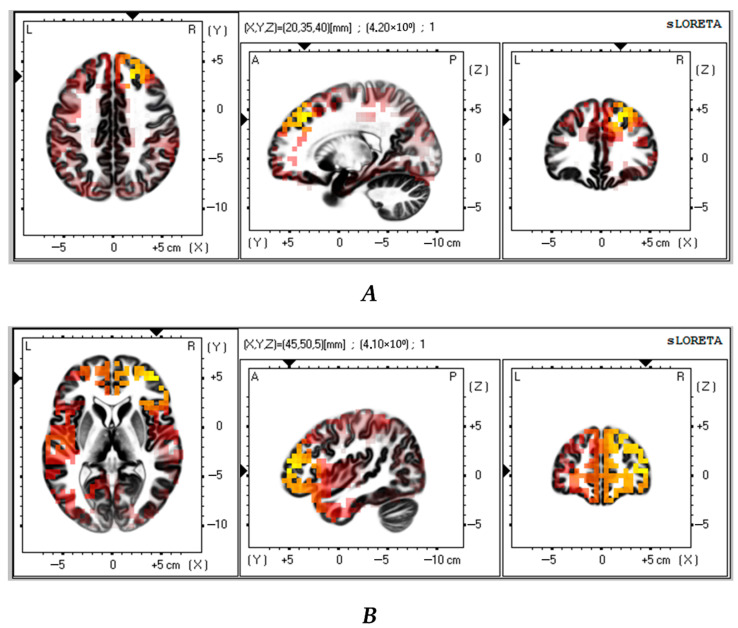
Source localization differences between successful VMT group and the control patients in theta (**A**) and alpha (**B**) activity (*n* = 23). Red and yellow blobs indicate that activity in VMT patients was lower than in the control patients.

**Table 1 biomedicines-13-01755-t001:** The baseline clinical and anamnestic characteristics of the virtual multitasking training (VMT) and control groups (*n* = 100).

Variable	VMT (*n* = 50)	Control (*n* = 50)	*p*-Value
Age, years, mean (SD)	62.5 (7.69)	62.9 (7.61)	0.77 *
Educational attainment, years, mean (SD)	12.9 (2.75)	13.1 (2.93)	0.54 *
MoCA, scores, mean (SD)	26.7 (2.13)	26.2 (2.29)	0.32 *
BDI-II, scores, mean (SD)	3.3 (3.1)	2.1 (2.04)	0.13 *
Smoking, *n* (%)	22 (44)	25 (50)	0.67 ^#^
Functional class NYHA, n (%)			0.55 ^#^
I–II	43 (86)	44 (88)	
III	7 (14)	9 (18)	
History of myocardial infarction, *n* (%)	22 (44)	29 (58)	0.09 ^#^
Fraction of left ventricle ejection, %, mean (SD)	61.6 (7.64)	58.4 (9.68)	0.31 *
Arterial hypertension, *n* (%)	42 (84)	43 (86)	0.55 ^#^
Type 2 diabetes mellitus, *n* (%)	11 (22)	13 (26)	0.61 ^#^
CA stenosis < 50%, *n* (%)	17 (34)	14 (28)	0.22 ^#^

BDI-II, Beck’s Depression Inventory; CA, carotid stenosis; MoCA, Montreal Cognitive Assessment; NYHA, heart failure according to the New York Heart Association. * between-group differences by Mann–Whitney U test; ^#^ between-group differences by *χ*^2^.

**Table 2 biomedicines-13-01755-t002:** The percentage of relative changes in postoperative indicators compared to baseline in the successful and unsuccessful groups of patients with virtual multitask training.

Cognitive Indicators	Patients with Successful VMT	Patients with Unsuccessful VMT	*p*
*Psychomotor and executive functions*			
** *Complex visual–motor reaction* **			
Reaction time, ms	4.53	7.62	0.32
Errors, *n*	23.62	−14.81	0.18
** *Level of functional mobility of nervous processes: responses to feedback* **			
Reaction time, ms	0.61	−6.07	0.01
Errors, *n*	−13.15	−9.38	0.68
Missed signals, *n*	5.10	2.64	0.76
*Attention*			
** *Bourdon’s test* **			
Processed letters per min, *n*	−14.83	−10.79	0.79
Processed letters per 4 min, *n*	−16.90	−74.78	0.36
Attention ratio, scores	−34.93	3.83	0.002
***Attention span test***, scores	−21.10	14.61	0.001
*Short-term memory*			
10-word memorizing test, *n*	−28.36	−18.10	0.03
10-number memorizing test, *n*	−23.26	1.79	0.63
Figurative memory, *n*	3.20	−4.26	0.44

**Table 3 biomedicines-13-01755-t003:** MNI coordinates for brain areas showing the largest CSD differences (top five) between the preoperative and postoperative alpha (7–9 Hz) values in patients with virtual multitask training (*n* = 50).

Rank	t-Value	MNI Coordinate			Brodmann Area	Brain Structure	*p*-Value
		X	Y	Z			
1	−3.83	20	−50	0	30	Limbic Lobe, Parahippocampal Gyrus	<0.05
2	−3.74	15	−45	0	30	Limbic Lobe, Parahippocampal Gyrus	<0.05
3	−3.73	10	−45	5	29	Limbic Lobe, Posterior Cingulate	<0.05
4	−3.69	20	−45	−5	19	Limbic Lobe, Parahippocampal Gyrus	<0.05
5	−3.68	25	−55	0	30	Limbic Lobe, Posterior Cingulate	<0.05

**Table 4 biomedicines-13-01755-t004:** MNI coordinates for brain areas showing the largest CSD differences (top five) between the preoperative and postoperative theta (3–5 Hz) and alpha (7–9 Hz) values in the control patients (*n* = 50).

Rank	t-Value	MNI Coordinate			Brodmann Area	Brain Structure	*p*-Value
		X	Y	Z			
*Theta-frequency band (3–5 Hz)*							
1	−3.42	−5	−75	10	23	Occipital Lobe, Cuneus	<0.05
2	−3.41	−5	−85	15	18	Occipital Lobe, Cuneus	<0.05
3	−3.40	0	−75	15	18	Occipital Lobe, Cuneus	<0.05
4	−3.38	−5	−80	10	17	Occipital Lobe, Cuneus	<0.05
5	−3.38	0	−80	15	18	Occipital Lobe, Cuneus	<0.05
*Alpha-frequency band (7–9 Hz)*							
1	−4.19	−5	−95	10	18	Occipital Lobe, Cuneus	<0.05
2	−4.17	−5	−100	10	18	Occipital Lobe, Middle Occipital Gyrus	<0.05
3	−4.15	−5	−90	10	18	Occipital Lobe, Cuneus	<0.05
4	−4.12	−5	−100	5	18	Occipital Lobe, Cuneus	<0.05
5	−4.10	0	−100	5	18	Occipital Lobe, Cuneus	<0.05

**Table 5 biomedicines-13-01755-t005:** MNI coordinates for brain areas showing the largest CSD differences (top five) between successful VMT group and control patients in the postoperative theta (3–5 Hz) and alpha (7–9 Hz) values (*n* = 23).

Rank	t-Value	MNI Coordinate			Brodmann Area	Brain Structure	*p*-Value
		X	Y	Z			
*Theta-frequency band (3–5 Hz)*							
1	3.89	20	30	45	8	Frontal Lobe, Middle Frontal Gyrus	<0.05
2	3.85	20	35	45	8	Frontal Lobe, Superior Frontal Gyrus	<0.05
3	3.83	−20	−65	15	31	Limbic Lobe, Posterior Cingulate	<0.05
4	3.80	25	30	45	8	Frontal Lobe, Middle Frontal Gyrus	<0.05
5	3.76	25	35	50	8	Frontal Lobe, Superior Frontal Gyrus	<0.05
*Alpha-frequency band (7–9 Hz)*							
1	3.62	10	55	35	9	Frontal Lobe, Superior Frontal Gyrus	<0.05
2	3.62	30	40	40	9	Frontal Lobe, Middle Frontal Gyrus	<0.05
3	3.61	10	50	40	9	Frontal Lobe, Medial Frontal Gyrus	<0.05
4	3.60	15	50	45	8	Frontal Lobe, Superior Frontal Gyrus	<0.05
5	3.60	35	40	15	10	Frontal Lobe, Middle Frontal Gyrus	<0.05

## Data Availability

The original contributions presented in the study are included in the article, further inquiries can be directed to the corresponding author.

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
