# Peer review of "Neurophysiological Effects of Virtual Reality Multitask Training in Cardiac Surgery Patients: A Study with Standardized Low-Resolution Electromagnetic Tomography (sLORETA)"

_biomedicines, 2025, doi:10.3390/biomedicines13071755_

Round 1
Reviewer 1 Report
Comments and Suggestions for Authors
The paper is solid and complete. Also, it is honest regarding the interpretation of obtained results and it acknowledges the study limitations. As advice for future research, I suggest the authors to also consider non-linear feature extraction from EEG, such as fractal dimension algorithms, among others.
I've found a set of issues that should be addressed before publishing:
- I've missed a expanded related work review, especially regarding other works evaluating different training approaches on brain after cardiac surgery.
- I've found some minor redaction errors in the text that should be checked before publishing.
- Figure 1 should specify that the timeline represents days.
- As I To ensure reproducibility, VR app should be available for future researchers in some way.
- MNI acronym is not defined
The paper is solid and complete. Also, it is honest regarding the interpretation of obtained results and it acknowledges the study limitations. As advice for future research, I suggest the authors to also consider non-linear feature extraction from EEG, such as fractal dimension algorithms, among others.
Author Response
POINT-BY-POINT RESPONSE TO THE REVIEWER COMMENTS
Dear Reviewer,
We would like to express our gratitude for your valuable comments and suggestions on our manuscript. Your insights have been instrumental in helping us to improve the quality of our study.
Reviewer 1
The paper is solid and complete. Also, it is honest regarding the interpretation of obtained results and it acknowledges the study limitations. As advice for future research, I suggest the authors to also consider non-linear feature extraction from EEG, such as fractal dimension algorithms, among others.
I've found a set of issues that should be addressed before publishing:
- I've missed a expanded related work review, especially regarding other works evaluating different training approaches on brain after cardiac surgery.
Response: Thank you for your contribution to the improvement of our manuscript. We corrected and expanded the Introduction section. Unfortunately, the effects of VR training on cognitive function in patients undergoing cardiac surgery have rarely been studied.
- I've found some minor redaction errors in the text that should be checked before publishing.
Response: We engaged the services of a professional translator in order to enhance the quality of our English.
- Figure 1 should specify that the timeline represents days.
Response: We corrected Figure 1.
- As I to ensure reproducibility, VR app should be available for future researchers in some way.
Response: At present, the documents are being processed in compliance with copyright law. In the future, the developed virtual reality (VR) application will be made available upon a special request. We previously published an article describing our VR application (Tarasova, I.; Trubnikova, O.; Kukhareva, I.; Kupriyanova, D.; Sosnina, A. The Neurophysiological Effects of Virtual Reality Application and Perspectives of Using for Multitasking Training in Cardiac Surgery Patients: Pilot Study. Appl. Sci. 2024, 14, 10893. https://doi.org/10.3390/app142310893).
- MNI acronym is not defined.
Response: We corrected this.

Reviewer 2 Report
Comments and Suggestions for Authors
This is a well-structured and timely article that investigates the neurophysiological effects of virtual reality-based multitask training (VMT) on cardiac surgery patients, using EEG and sLORETA. The authors target an important clinical problem—postoperative cognitive dysfunction (POCD)—and propose an innovative non-pharmacological intervention. The manuscript integrates clinical, cognitive, and electrophysiological assessments, and the methodology is mostly sound. However, several areas require clarification, refinement, and critical elaboration to reach the standards of a high-impact publication.
Add more emphasis on the translational potential in the conclusion of the abstract.
The rationale for using multitask training vs. single-task or cognitive-only VR interventions needs clearer justification. The neurophysiological hypothesis could be more explicitly linked to existing models of neural plasticity or cortical reorganization. I suggest adding 1–2 sentences linking VMT effects to neural plasticity frameworks.
Please, state explicitly whether the randomization process was blinded or concealed. Also, justify why only male participants were included.
Clarify how adherence to the training protocol was ensured (did all patients complete the same number of sessions?).
Could the removal of key central channels (Fz, Cz, Pz) may have reduced the ability to detect midline activation? Explain why these channels were excluded and how this may have affected source localization.
The response rate of 46% success (23 of 50 VMT participants) is low, and further discussion is needed. Intergroup comparisons rely on a subset (23) vs. full control group. This could introduce selection bias. I suggest to include intention-to-treat and per-protocol comparisons to support the robustness of your key findings.
The discussion of "successful VMT" vs. "non-successful" lacks depth. What distinguished these patients? The draft does not explore whether these EEG changes are transient or predictive of long-term outcomes. I suggest you to explore possible individual predictors of VMT responsiveness. You should recommend follow-up studies or integration with neuropsychological test gains.
The report missed some limitations: no long-term follow-up on cognitive recovery; EEG markers were not validated against behavioral or clinical outcomes. Please, acknowledge all limitations.
Figure captions should specify the meaning of red vs. blue. Consider adding behavioral results (e.g., MoCA, POCD rates) in graphics for easier comparison.
A few references are cited in duplicate (e.g., [21] appears in both VR setup and EEG references). Consider differentiating them.
Comments on the Quality of English LanguageDear Authors,
The manuscript is scientifically readable and mostly clear, but it exhibits numerous stylistic inconsistencies, awkward constructions, and grammatical errors that detract from its professional polish. It would benefit from a thorough language revision by a native or expert science editor.
Frequent Issues:
Article usage (a/an/the):
Examples:
“...a rise human life expectancy.” → should be “a rise in human life expectancy.”
“...the patient was driving a tractor and observed a road...” → should be “the patient drove a tractor and observed a road...” (past continuous tense would be more fluent here).
Verb tense consistency:
Present and past tense are sometimes mixed incorrectly.
Example: “It is known that ischemic brain damage... has a multifocal localization.” better: “is known to have”.
Subject-verb agreement:
Example: “...and the threshold value for cognitive decline was equal to 20%” grammatically correct, but “was set at 20%” would be more fluent.
Redundancy:
Phrases like “...before CABG surgery” (CABG already implies surgery) could be simplified.
The term “the patient was instructed to...” is repeated with similar structure in close succession.
The manuscript maintains a mostly academic tone appropriate for Biomedicines.
Technical terms are well-defined (e.g., sLORETA, CSD, POCD).
Long, convoluted sentences hinder readability.
“According to the hypothesis, VR technologies and multitasking can enhance cognitive resources by using more brain tissue to perform different tasks.”
Here is a suggestion: “We hypothesize that VR multitasking enhances cognitive resources by engaging multiple brain regions simultaneously.”
While common in scientific writing, it is excessively used here, making the text less dynamic.
Example: “All subjects were treated in strict compliance...” could be revised as “We treated all subjects in compliance...”
Ambiguity in key phrases:
“Success of the VMT program was defined by the absence of POCD...” could be more clearly expressed: “We considered VMT successful if POCD was no longer present at follow-up.”
Overly technical phrasing without clarification:
Example: “Three orthogonal dipole moments...” — useful to briefly remind readers what these represent or how they contribute to source estimation.
Inconsistent abbreviation usage:
“CABG”, “VMT”, “POCD”, and “CSD” are sometimes reintroduced unnecessarily or used before being defined.
Ensure consistent formatting (e.g., “theta rhythm” vs. “theta-band activity”).
Improper punctuation with abbreviations:
There are places where punctuation is omitted after full terms or abbreviations (e.g., “CABG (control group)” lacks the article “the”).
Comma splices and run-on sentences:
Several sentences are improperly joined by commas instead of conjunctions or full stops.
Example: “It is also observed to have beneficial effects on psychological rehabilitation, such as decreasing stress...” This can be split or clarified with semicolons or conjunctions.
Hyphenation issues:
“multi-tasking” and “multitask” appear inconsistently — standardize usage to “multitask” per modern academic convention.
“Non-pulsatile”, “postoperative”, and “resting-state” are sometimes inconsistently hyphenated.
Author Response
POINT-BY-POINT RESPONSE TO THE REVIEWER COMMENTS
Dear Reviewer,
We would like to thank you for your valuable feedback on our manuscript. We appreciate your insights, which have been instrumental in assisting us with improving the quality of our research.
Reviewer 2
This is a well-structured and timely article that investigates the neurophysiological effects of virtual reality-based multitask training (VMT) on cardiac surgery patients, using EEG and sLORETA. The authors target an important clinical problem—postoperative cognitive dysfunction (POCD)—and propose an innovative non-pharmacological intervention. The manuscript integrates clinical, cognitive, and electrophysiological assessments, and the methodology is mostly sound.
However, several areas require clarification, refinement, and critical elaboration to reach the standards of a high-impact publication.
- Add more emphasis on the translational potential in the conclusion of the abstract.
Response: Thank you very much for your comment. We corrected the Conclusion section of the abstract.
- The rationale for using multitask training vs. single-task or cognitive-only VR interventions needs clearer justification. The neurophysiological hypothesis could be more explicitly linked to existing models of neural plasticity or cortical reorganization. I suggest adding 1–2 sentences linking VMT effects to neural plasticity frameworks.
Response: We have revised and expanded the Introduction section of the manuscript.
- Please, state explicitly whether the randomization process was blinded or concealed. Also, justify why only male participants were included.
Response: A pseudo-randomization method was used to form two groups, in dependence on the baseline clinical and anamnestic characteristics. Only male participants were included in the study to eliminate the effect of sex differences on clinical, demographic characteristics, and EEG data. We added this information into section 2.1. Data Collection and Sampling of the manuscript.
- Clarify how adherence to the training protocol was ensured (did all patients complete the same number of sessions?).
Response: As listed in section 2.3 of the manuscript (Virtual reality-based multitask training paradigm), the VMT course was started 3-4 days after CABG and was conducted daily training sessions depending on the length of a particular patient's stay at the hospital stage (approximately 5-9 training sessions). Therefore, each participant completed at least 5 training sessions prior to discharge from the clinic. The mean number of training sessions was 6.7. We included this information in section 2.3 of the manuscript.
- Could the removal of key central channels (Fz, Cz, Pz) may have reduced the ability to detect midline activation? Explain why these channels were excluded and how this may have affected source localization.
Response: We would like to apologize for our mistake. We used the LORETA transformation matrix consisted of 60 channels. We did not remove the central channels (Fz, Cz, and Pz), we have removed two unnecessary channels (CB1 and CB2).
- The response rate of 46% success (23 of 50 VMT participants) is low, and further discussion is needed. Intergroup comparisons rely on a subset (23) vs. full control group. This could introduce selection bias. I suggest to include intention-to-treat and per-protocol comparisons to support the robustness of your key findings.
Response: A preliminary statistical analysis of preoperative and postoperative data showed that there is no significant difference between CSDs in theta and alpha frequency bands in VMT patients before and after CABG compared to the full control group. We agree with the honorable reviewer that the results comparing successful VMT patients (n = 23) with controls (n =23) suggest lower reliability of these findings. A subgroup of control patients was selected randomly (every second patient), but it should be noted that the LORETA analysis results are a pilot study. In the future, we intend to conduct follow-up studies with a larger sample size.
- The discussion of "successful VMT" vs. "non-successful" lacks depth. What distinguished these patients? The draft does not explore whether these EEG changes are transient or predictive of long-term outcomes. I suggest you to explore possible individual predictors of VMT responsiveness. You should recommend follow-up studies or integration with neuropsychological test gains.
Response: We would like to express our gratitude to the honorable reviewer for the comments. We discussed it in more detail. The behavioral findings of our research (Table 2) demonstrate that patients who achieved successful outcomes showed improvements in attentional and short-term memory dynamics.
- The report missed some limitations: no long-term follow-up on cognitive recovery; EEG markers were not validated against behavioral or clinical outcomes. Please, acknowledge all limitations.
Response: We have revised and expanded the Limitations section of the manuscript. Also, we would like to point out that we validated the LORETA data against behavioral and clinical outcome, such as POCD, in our study. To present the behavioral outcomes, we added an additional table (Table 2).
- Figure captions should specify the meaning of red vs. blue. Consider adding behavioral results (e.g., MoCA, POCD rates) in graphics for easier comparison.
Response: To each figure, we added a legend that denotes blue and red blobs. To present the behavioral outcomes, we added an additional table (Table 2).
- A few references are cited in duplicate (e.g., [21] appears in both VR setup and EEG references). Consider differentiating them.
Response: We corrected the references list.
Comments on the Quality of English Language
Response: We appreciate the time and effort that you have dedicated to providing your valuable feedback on our manuscript. We have incorporated changes to reflect your suggestions.
We engaged the services of a qualified translator to review the manuscript.

Round 2
Reviewer 2 Report
Comments and Suggestions for Authors
Dear authors,
Thank you for your effort to improve the manuscript's readability and clarity.
I have no further comments to add.
Regards.